# Automatic Myocardium Segmentation in Arterial Spin Labeling Perfusion MRI Using Uncertainty-Aware Mask R-CNN

**Anne Oyarzun-Domeño**[1,2]                              ANNE.OYARZUN@UNAVARRA.ES
[1] *Electrical Electronics and Communications Engineering, Public University of Navarre, 31006, Pamplona, Spain.*
[2] *IdiSNA, Health Research Institute of Navarra, 31008, Spain.*

**Verónica Aramendía-Vidaurreta**[2,3]                    VARAMENDIA@UNAV.ES
[3] *Department of Radiology, Clínica Universidad de Navarra, 31008, Pamplona, Spain.*

**Gorka Bastarrika**[2,3]                                 BASTARRIKA@UNAV.ES

**Ana Ezponda**[2,3]                                      AEZPONDA@UNAV.ES

**María A. Fernández-Seara**[2,3]                         MFSEARA@UNAV.ES

**Arantxa Villanueva**[1,2,5]                             AVILLA@UNAVARRA.ES
[5] *Institute of Smart Cities (ISC), Health Research Institute of Navarra, 31006, Pamplona, Spain.*

## Abstract

Coronary artery disease (CAD) is a leading cause of cardiovascular morbidity and mortality worldwide. Assessing myocardial perfusion is important to detect potential areas of ischemia in patients with suspected CAD. Arterial spin labeling (ASL) allows non-invasive quantification of myocardial perfusion using arterial blood as endogenous tracer. Segmentation of the left ventricular myocardium is critical in the post-processing for ASL images, but it is challenging due to low signal-to-noise ratio (SNR). This study introduces an automatic myocardium segmentation pipeline including uncertainty awareness, employing Mask R-CNN with dropout layers to capture model uncertainty. Our dataset consists of flow-sensitive alternating inversion recovery (FAIR) ASL images from 16 patients with suspected CAD. Our approach achieves robust segmentation results, with similarity coefficient of 75% and 0.3% misclassification rate. We obtain an 80% correlation with real perfusion values.

**Keywords:** Mask R-CNN, uncertainty, dropout, Myocardial Blood Perfusion

## 1. Introduction

CAD refers to the obstruction of the coronary arteries, the vessels responsible for delivering oxygenated blood to the heart. Early detection of CAD involves assessing myocardial perfusion using magnetic resonance imaging (MRI). ASL is a non-invasive MRI method that enables the quantification of myocardial blood flow (MBF) and involves obtaining two distinct images: one where the magnetization of water protons in the blood is inverted (label, L) and another where magnetization of blood perfusing the myocardium is not inverted (control, C) (Detre et al., 1994). These images are then subtracted to generate perfusion-weighted images, where the differences in contrast within the myocardium are directly proportional to MBF. Given the relatively low perfusion signal, multiple acquisitions are necessary to enhance the SNR. Thus, the segmentation of the left ventricular myocardium is an essential component of the post-processing workflow, as it enables the assessment of global perfusion measurements (Cerqueira et al., 2002). In this work, MBF is estimated

using the following equation derived from Buxton's general kinetic model (Buxton et al., 1998):

$$MBF(ml/g/min) = \frac{\lambda \cdot (C - L)}{2 \cdot B \cdot TI \cdot e^{\frac{-TI}{T1}}} \tag{1}$$

where $C - L$ denotes the average myocardium ASL signal difference. Outliers are discarded if the mean value is 2 standard deviation (SD) away from the global mean. $\lambda$ refers to the blood/tissue water partition coefficient (1 mL/g), $B$ represents the mean intensity (a.u.) of the myocardium in the baseline image, $TI$ stands for the inversion time (1000 ms), and $T_1$ is the relaxation time of blood at 1.5 Tesla (1434 ms). **Aim**. We propose the following pipeline for myocardium segmentation in FAIR-ASL: 1) to implement Mask R-CNN for the automatic segmentation of myocardium, 2) to apply and enable dropout layers in training and inference, respectively, to study the uncertainty of model, 3) to automatically estimate the MBF based on predicted segmentation masks. Our work presents advantages over semi-automated myocardial segmentation methods (Aramendía-Vidaurreta et al., 2021) and shows results similar to those reported in the literature (Do et al., 2020).

## 2. Materials and Methods

**Dataset** consists of FAIR rest-stress cardiac ASL images from a previously published study (Aramendía-Vidaurreta et al., 2021) with 16 patients with suspected CAD in which segmentation masks were available and are used as ground truth (GT) in this work. Images were acquired in a mid-ventricular short axis slice of the myocardium, with matrix size of 128 x 104. A baseline image $B$ without presaturation and inversion pulses was acquired to allow MBF quantification. 26 stress and 48 rest scans were acquired per patient.

**Architecture**. Mask R-CNN operates as a three-stage object detection system: a region proposal network (RPN), a region-based convolutional neural network (RCNN), and a semantic segmentation model (He et al., 2018).

**Preprocessing**. Intensity-windowing (15%) is used for contrast enhancing in training and inference stages. The model is trained on 838 images, validated on 236, and tested on 148, belonging to 11, 3, and 2 subjects, respectively.

**Training**. We select the Resnet101 as the backbone of Mask R-CNN and feature pyramid network (FPN) to improve the quality of feature maps used for both bounding box regression and mask prediction. We implement a dropout layer at the end of each block of the backbone and after the activation layer of the FPN. The model is trained for 200 epochs using a supervised gradient descent optimizer and learning rate of $10^{-3}$. During training, data augmentation (rotations and translations) is applied to image/mask pairs which are resized to dimensions of 128 x 128 using zero-padding. We use Python 3.7 and TensorFlow on GPU NVIDIA GeForce RTX 3060 for model training and testing. Training takes $\approx$ 240 min.

**Inference**. As stated in (Gal and Ghahramani, 2016), dropout has been shown to serve as a Bayesian approximation, offering model uncertainty through Monte Carlo (MC) dropout during testing. In (Do et al., 2020), an MC dropout U-Net was used for segmentation of myocardium in ASL and assessed model uncertainty. In this work, dropout layers are enabled during inference, and we sample multiple predictions for each image by repeating the forward passes for the same input image and model $n = 100$ times. Because of the

inherent randomness in MC dropout, predictions undergo slight variations, allowing us to sample the distribution of model outputs. Each forward pass takes 0.05 seconds/image. The mean and SD of $n = 100$ predicted masks are used to generate the final predicted segmentation and the pixel-by-pixel uncertainty map for each test image.

**Evaluation.** We evaluate the performance of our segmentation approach considering various metrics such as true positive (sensitivity) and negative rates (specificity), spatial overlap (dice coefficient, DSC), and misclassification rate (false positive rate, FPR). Furthermore, we utilize Pearson's correlation coefficient to assess the correlation between MBF values computed with the GT masks and those derived from predicted myocardium masks. Finally, qualitative evaluation is made on uncertainty maps (see Figure 1).

## 3. Results and discussion

The results of the study indicate that the median values $\pm$ SD for FPR, sensitivity, and specificity were $0.003 \pm 0.000$, $0.798 \pm 0.034$, and $0.994 \pm 0.001$, respectively, while the median value for the DSC was $0.748 \pm 0.036$. These findings suggest a high specificity and sensitivity of the method, with a low false positive rate. We paid particular attention to low FPR values to mitigate the impact of partial volume, thereby preventing the interference of signals originating from ventricular blood pools and/or epicardial fat (Do et al., 2020). Moreover, the low SD values of all the scores indicate robustness and stability in the model's performance. The concordance correlation coefficient was 0.810 between MBF measured using the automatic segmentation mask with that measured using GT segmentation masks. Figure 1 depicts segmentation examples on C and L images alongside with corresponding uncertainty maps, that indicate increased uncertainty along the outer boundaries of the myocardium mask. Further investigation may be warranted to explore the variability observed in the DSC, which could potentially affect its reliability in certain applications. Additionally, assessing the performance of the method across different datasets or conditions could provide valuable insights into its robustness and generalizability. In general, our model demonstrates efficient automatic myocardium segmentation.

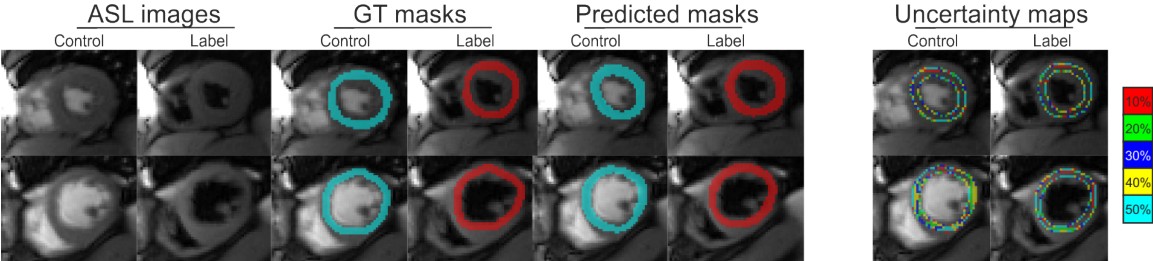

Figure 1: Examples of GT and automatic myocardium segmentation masks. Uncertainty maps (on the right) represent the uncertainty levels (%) associated with each segmentation across $n = 100$ forward passes in the inference stage.

## Acknowledgments

Anne Oyarzun-Domeño received a predoctoral grant No. 0011-0537-2021-000050 from the Department of University, Innovation, and Digital Transformation, Government of Navarra. Verónica Aramendía-Vidaurreta received grant support from Siemens Healthcare Spain. We acknowledge grant No. PI21/00578 from the Spanish Ministry of Science and Innovation.

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
