# OpenReview forum: "Automatic Myocardium Segmentation in Arterial Spin Labeling Perfusion MRI Using Uncertainty-Aware Mask R-CNN"
_MIDL.io/2024/Short_Papers — MIDL 2024 Short Papers_

### Official Review · Reviewer_J8tT · 2024-04-24

**Confidence:** 4
**Final Rating:** 4

**Review:**

This paper presents a pipeline for automatic myocardium segmentation on ASL (Arterial spin labelling) images. Mask R-CNN is used for the segmentation and dropout layers are added to obtain model uncertainty. Uncertainty increases along the boundaries. The method is evaluated on a dataset with 16 patients.

Strength:
- Well-written short paper
- pipeline is well-explained
- proposed uncertainty (even if there is no particular use of this uncertainty after)

Weakness :
- only 16 patients
- comparison with other existing segmentation methods is missing

---

### Decision · Program_Chairs · 2024-04-26

Accept